# Immunomodulatory, Anticancer, and Antimicrobial Effects of Rice Bran Grown in Iraq: An In Vitro and In Vivo Study

**DOI:** 10.3390/ph15121502

**Published:** 2022-12-01

**Authors:** Wamidh H. Talib, Asma Ismail Mahmod, Dima Awajan, Reem Ali Hamed, Intisar Hadi Al-Yasari

**Affiliations:** 1Faculty of Allied Medical Sciences, Applied Science Private University, Amman 11931-166, Jordan; 2Department of Clinical Pharmacy and Therapeutic, Applied Science Private University, Amman 11931-166, Jordan; 3Department of Genetic Engineering, College of Biotechnology, Al-Qasim Green University, Babylon 964, Iraq

**Keywords:** rice bran, immune modulation, anticancer, natural products, animal study

## Abstract

Emerging evidence supports the role of rice bran in cancer prevention. Studies were conducted on multiple rice cultivars. However, limited studies were conducted on rice cultivars in the Middle East. In this study, rice bran growing in Iraq (*O. sativa* ssp. Japonica, cultivars: Amber Barka) was evaluated for its effect on preventing cancer and stimulating the immune system. Rice bran was collected from local mills in Al-Najaf (south of Iraq). Several solvent extracts (ethanol, methanol, n-hexane, and water) were prepared by maceration. MTT assay was used to measure the antiproliferative effects of extracts against a panel of cancer cell lines. The ability of each extract to induce apoptosis and inhibit angiogenesis was measured using standard ELISA kits. The effect of extracts on the immune system was evaluated using a lymphocyte proliferation assay, a pinocytic activity assay, a phagocytic activity assay, and a Th1/Th2 cytokine detection kit. A microbroth dilution method was used to detect the antimicrobial activity of each extract against different microbial strains. LC–MS analysis was used to detect the phytochemical composition of extracts, while DPPH assay was used to determine the antioxidant activity. For the in vivo study, rice bran was added to mouse fodder at 10% and 20%. Mice were treated for two weeks using mouse fodder supplemented with rice bran. In the third week of the experiment, EMT6/P breast cancer cells (1 × 10⁶ cells/mL) were injected subcutaneously into the abdominal area of each mouse. The dimensions of the grown tumors were measured after 14 days of tumor inoculation. A microbroth dilution method was used to evaluate the antimicrobial activity of rice bran extracts against three bacterial strains. The highest antiproliferative activity was observed in ethanol and *n*-hexane extracts. Ethanol and methanol extract showed the highest activity to induce apoptosis and inhibit angiogenesis. Both extracts were also effective to enhance immunity by activating lymphocytes and phagocytes proliferation with modulations of cytokine levels. The incorporation of rice bran in mice food caused a 20% regression in tumor development and growth compared with the negative control. All extracts exhibited limited antimicrobial activity against tested microorganisms. Methanol extract showed antioxidant activity with an IC_50_ value of 114 µg/mL. LC–MS analysis revealed the presence of multiple phytochemicals in rice bran including apiin, ferulic acid, and succinic acid. Rice bran is a rich source of active phytochemicals that may inhibit cancer and stimulate the immune system. Rice bran’s biological activities could be due to the presence of multiple synergistically active phytochemicals. Further studies are needed to understand the exact mechanisms of action of rice bran.

## 1. Introduction

Among the most spreadable diseases in the world is still cancer. By 2020, there were 19.3 million new cases and approximately 10 million deaths [1]. Additionally, as predicted by the World Health Organization, more than 20 million new cancer cases will arise before 2025, specifically in low- and middle-income countries which include the middle east and north Africa (MENA) region [2,3]. This is due to limited resources in health care services, lack of early detection programs for cancer, and poor therapy approaches [4,5]. For instance, in Jordan, it is considered the second primary reason for mortality after cardiovascular diseases, exceeding breast cancer as the most prevalent malignancy [6,7]. Additionally, 33,661 cases of cancer were reported from 1996 to 2005 [8]. The number of cancer cases also increased by 44% over ten years, as cases increased from 3362 to 4849 (2000–2010) [6]. Additionally, a study performed among Syrian refugees in Jordan reported that 869 Syrians are diagnosed with cancer in Jordan yearly and the cost of the treatment was 15.6 million Jordan dinars (USD 22.1 million) [9]. Cancer treatment approaches include chemotherapy, surgery, and irradiation while chemotherapy is the major component of the treatment regimen. However, its usefulness is restricted because of its non-selectivity toward cancer cells over non-cancerous cells, leading to ineffective therapy. In addition, drug resistance against chemotherapeutic agents also affects the treatment process [10].

Due to these statistics, economical issues, and side effects of chemotherapy, it is important to keep discovering new anticancer agents. Cancer is a multifactorial disease triggered by the hybrid effects of both genetic and environmental factors [11]. Mucci et al. estimated that 33% of cancer cases are related to genetic factors [12]. So far, a great number of genetic loci have been recognized as vulnerability markers for definite cancers by genome-wide association studies (GWAS) [13].

On the other hand, several environmental factors affect cancer development such as outdoor air pollution, exposure to tobacco smoke, radon exposure, inorganic arsenic in drinking water, water chlorination by-products, sunlight exposure, and diet [14,15,16]. Over 40% of all cancer cases and losses are accountable to possibly adaptable risk factors, mainly unhealthy lifestyles [17].

Diet is considered a significant risk factor for many diseases and premature death; fortunately, it is a modifiable risk factor that we could control and alter, thus improving many conditions such as hypertension, diabetes mellitus, cancer, and obesity. The Mediterranean diet is a generic name for the traditional dietary patterns of the people residing in the Mediterranean region [18] and it is considered among the most nutritious dietary patterns globally, it involves fish, monounsaturated fats from olive oil, fruits, vegetables, whole grains, legumes/nuts, and moderate alcohol consumption [19]. Rice bran is among the ingredients of the Mediterranean diet. Rice bran contains necessary fatty acids, dietary fibers, proteins, dietary minerals, and vitamins [20]. It has been reported to have an active role in many diseases and pathological conditions, such as diabetes mellitus and cardiovascular diseases, as bran extract has a role in lowering cholesterol [21] and glucose levels [22]. Further, it has multiple roles as an anticancer—it reduces inflammation, induces cell arrest, promotes cell apoptosis, and enhances the chemotherapeutic effect [23]. Kannan et al. reported that a pentapeptide isolated from rice bran inhibited cancer proliferation in colon, breast, lung, and liver cancer cells [24]. Additionally, it has a role in immunotherapy as it activates the natural killer cells [25]. 

Phytochemical analysis showed the presence of extensive genetic variation among rice cultivars [26]. Rice bran metabolite profiling of three USA cultivars revealed appreciable diversity in bioactive compounds [27]. The diversity in phytochemical composition is linked to genetic and environmental factors. Stressors such as drought, pesticides, and nutrient limitation were reported as inducers for phytochemical heterogeneity among different rice cultivars [28].

Many findings reported the immunomodulatory and therapeutic activities of rice bran, but this is the first study that investigates the immunomodulatory and anticancer activities of rice bran cultivated in Iraq.

Iraq has a unique environment with temperatures exceeding 50 °C in summer. Drought is also a serious problem in Iraq that affects all crops including rice [29]. However, there are no studies providing details about the immunomodulatory, anticancer, and antimicrobial activities of rice bran growing in a unique environment such as Iraq.

In this study, bran from Iraqi rice cultivars was subjected to phytochemical analysis followed by evaluation of anticancer, immunomodulatory and antimicrobial effects. 

## 2. Results

### 2.1. LC–MS Analysis of Rice Bran Extracts

The results of LC–MS analysis exhibited the presence of different phytochemicals that may play a part in rice bran extract biological activity. Apiin, a natural flavonoid, was found in high percentages in rice bran extracts, including methanol extract (31.1%), ethanol extract (15.9%), and aqueous extract (10.1%). Further, methanol extract showed a high content of ferulic acid and 5,6,4′-trihydroxy-7,3′-dimethoxyflavone with a relative percentage of 13.7% and 8%, respectively. Succinic acid was the main natural compound present in aqueous extract with a relative percentage of 70.2%. On the other hand, *n*-hexane and ethanol extracts showed a high existence of bioactive lipids known as FAHFA, which stand for fatty acyl esters of hydroxy fatty acid (Table 1).

### 2.2. The Antiproliferative Activity of Rice Bran Extracts

Based on the MTT results, ethanol and *n*-hexane extracts exhibited antiproliferation activity against the MCF-7 cell line, with IC_50_ values of 0.36 and 0.12, respectively. Further, methanol extract showed a cytotoxic effect toward T47D and EMT6/P cells, with IC_50_ values of 1.01 and 1.19, respectively (Table 2). On the other hand, aqueous extract was less effective and showed a percentage of cell survival greater than the other tested extracts (Figure 1).

### 2.3. Apoptotic Activity of Rice Bran Extracts against the T47D Cell Line

A high level of caspase-3 is a signal of promoting apoptosis and reducing cell growth. Caspase-3 assay results showed the activity of ethanol and methanol extracts through inducing caspase-3 level by 5.4 and 3.1 fold compared to the negative control. However, *n*-hexane extract exhibited no activity compared to the negative control (Figure 2).

### 2.4. The Effect of Rice Bran Extracts on VEGF Expression in T47D Cells

VEGF expression was detected in vitro in T47D cells to describe the mechanism of the antiproliferative activity of bran extracts. Methanol and ethanol extracts were able to reduce the concentration of VEGF (200 and 241 pg/mL), respectively, compared to the negative control (366 pg/mL). On the other hand, n-hexane extract was less active with VEGF expression close to the negative control (Figure 3).

### 2.5. Effect of Rice Bran Consumption on Tumor Development and Growth

An in vivo study was designed with two stages of treatment protocol; the prophylactic stage, which involved feeding the mice fodder mixed with specific percentages of rice bran for 14 days, followed by the treatment stage after tumor inoculation with the same feeding conditions. After another 14 days, the mice were sacrificed and tumors were isolated, and measurements of their dimensions and weight were taken using caliper digital. Based on the results, there was a reduction in tumor size in both bran groups (10% rice bran and 20% rice bran), with a percentage of change of 50.1% and 82.6%, respectively, compared to the negative control (133.2%) (Table 3). The average tumor size showed a reduction in tumor development compared to the control (Figure 4). The percentage of mice with no detectable tumor for 10% rice bran and 20% rice bran was 60% and 70%, respectively, while control group was 50% (Table 3 and Figure 5).

### 2.6. The Effect of Rice Bran Extracts on the Proliferation of Splenic Lymphocytes 

In the presence of co-mitogens (Con A or LPS), methanol extract exhibited the highest stimulation index (4.3) at a concentration of 5 mg/mL (Figure 6A), followed by ethanol extract and *n*-hexane (Figure 6A,B). On the other hand, without co-mitogens, splenic lymphocytes were stimulated by ethanol extract (3.7) and methanol extract (3.6) at the highest tested concentration (5 mg/mL) (Figure 6C).

### 2.7. The Effect of Rice Bran Extracts on the Activity of Mouse Peritoneal Macrophages

In this experiment, the results demonstrated an improvement in the phagocytic effect of murine macrophages. Ethanol extract was able to stimulate macrophage activity with an index value of 133, followed by *n*-hexane (76) at a concentration of 5 mg/mL (Figure 7).

### 2.8. The Effect of Rice Bran Extracts on Cytokines of Murine Splenic Lymphocytes 

Different cytokines (IL-2, IL-4, IL-10, IFN-γ) were determined using TH1/TH2 assay kit. Briefly, *n*-hexane and methanol extracts were able to reduce IL-4 levels in lymphocytes treated with 5 mg/mL compared to the negative control (Figure 8B). However, methanol extract showed a slight modulation of IL-2 expression compared to the control (Figure 8C).

### 2.9. Antibacterial Activity of Rice Bran Extracts

The MIC was investigated based on the microtiter dilution method. According to the results, ethanol and *n*-hexane extracts were able to reduce the bacterial growth of *B. subtilis* at a concentration of 150 mg/mL (Table 4). Further, ethanol and methanol extracts showed antibacterial activity against *E. coli* with an MIC value of 150 mg/mL. However, all the extracts exhibited no activity against *P. aeruginosa* at the highest selected concentration.

### 2.10. Antioxidant Activity of Rice Bran Extracts 

The result of this assay is based on the ability of the tested samples to reduce the stable purple-colored radical DPPH to the yellow-colored diphenylpricrylhydrazine. Methanol and ethanol extracts have shown antioxidant activity, with IC_50_ values of 114 and 168 µg/mL, respectively (Table 5). Meanwhile, the aqueous extract was less active, with 48% scavenging potential at the highest tested concentration (400 µg/mL) (Figure 9). On the other hand, the n-hexane extract showed weak activity compared to the other extracts and the standard reference (IC_50_ of ascorbic acid was 1.74 µg/mL). 

## 3. Discussion

### 3.1. Anticancer Effect 

Using complementary and alternative medicine (CAM) in cancer treatment is very common, and involves the use of many agents including antioxidants, fibers, vitamins, and herbs [30]. These agents have a known mechanism of action, the capacity to be consumed orally, little to no toxicity, high efficacy across various sites, low cost, and most importantly, widespread acceptability [31]. Many cancer patients start using these products extensively; in a study conducted among patients taking chemotherapy in phase I trials, 88.2% of responders used at least one CAM modality [32]. In another study, it was also reported that using CAM was substantially higher among patients with cancer (32%), and specifically among cancer outpatients (50%), than among those with acute or long-term non-malignant disorders [33]. Many studies showed that using natural products along with cancer treatment enhanced cancer treatment and decreased the side effects of many existing therapies. In this research, crude extracts of rice bran were prepared and then evaluated for their anticancer, antibacterial and immunomodulatory effects. Different extracts were prepared by maceration, then assessed for their antiproliferative activity against MCF-7, T47D, MDA-MB-231, and EMT6/P. An in vivo study was conducted to observe the prophylactic and antitumor properties of the rice bran on female Balb/C mice inoculated with EMT6/P mouse breast cancer. Rice bran extracts were efficient in impeding tumor progression in both in vitro and in vivo experiments. Ethanol and *n*-hexane extracts demonstrated antiproliferation action against MCF-7 cells while methanol extract was active against T47D and EMT6/P cells. On the other hand, the aqueous extract less active and showed a higher percentage of cell survival.

The active phytochemical components were identified as LC–MS analysis was performed on all extracts. Methanol extracts mainly had flavonoids, phenolic compounds, and flavones, though hexane extract had mainly fatty acyl esters of hydroxy fatty acid. On the other hand, ethanol extract was rich in flavonoids and fatty acyl esters of hydroxy fatty acid. These results are consistent with other phytochemical content analyses performed on rice bran from Indonesia which revealed that this bran is rich in phenolic, flavonoid, alkaloid, triterpenoid, steroid, and saponin compounds [34]. The results of LC–MS analysis indicated that alcoholic extracts are full of phenolic components such as apiin, ferulic acid(trans),5,6,4′-trihydroxy-7,3′-dimethoxyflavone,3,5-dimethoxy-4-hydroxyacetophenone, p-Coumaric acid, saponarin, and rutin. Oki et al. noticed that phenolic content and free-radical scavenging activity were positively correlated [35]. Therefore, bran extract may reduce the amount of cancer-causing reactive oxygen species (ROS) in cancer cells. Additionally, phenolic compounds could inhibit the vascular endothelial growth factor which has a key role in cancer development, and they could inhibit the proliferation of cancer cells [36]. 

Bran alcoholic extracts were also rich in flavonoids such as apiin, 5,6,4′-trihydroxy-7,3′-dimethoxyflavone, saponarin, and rutin. Previous studies showed that flavonoids exhibited extensive anticancer activity against MCF-7, HeLa, and Hep2 cancer cells [37], also flavonoids revealed an apoptotic effect in different cancer cells [38,39,40], moreover they inhibit tumor cell invasion [41]. Additionally, in another study, it was found that flavonoids trigger cell cycle arrest and induce apoptosis [42]. Likewise, flavonoids impeded the catalytic activity of topoisomerase I, thus interrupting DNA replication in cancer cells, resulting in cancer cell death [43]. In addition, it was found that flavonoids, especially those which have a double bond in the second ring structure such as kaempferol, apigenin, quercetin, and rutin showed strong inhibition of cell proliferation and VEGF expression in human ovarian cancer cells [44].

In a study, it was found that fatty acid esters of hydroxy fatty acids may be a protective mechanism for cancer cells against apoptosis [45]. Such results may explain why hexane extract is not as effective as menthol and ethanol extract in inducing apoptosis and inhibiting VEGF, as hexane extract had the lowest phenolic compounds among these extracts. Additionally, it has the highest fatty acid esters of hydroxy fatty acids, which can be a protective mechanism against apoptosis. Further, this explains why alcoholic extracts are the most effective at inhibiting VEGF and inducing apoptosis as they were the richest in phenolic compounds.

Both methanol and ethanol extracts were rich in Apiin, which is a diglycoside of the flavone apigenin. Apigenin showed an ability to inhibit ER-positive breast cancer cell proliferation through the Akt/FOXM1 signaling pathway [46]. In addition, apigenin stimulated cell cycle arrest and apoptosis in oral squamous cell carcinoma in vitro [47]. The presence of this flavonoid supports the apoptotic and antiproliferative effect of the bran extract results obtained in our study.

Saponarin, which is another natural flavonoid found in rice bran extract, has a strong antioxidant effect [48]. This may be among the mechanisms through which bran extract fights cancer.

Another compound detected in rice bran extract was rutin, which is a flavone, and inhibits A549 lung cancer cells and colon cancer cells effectively [49]. Additionally, Maeda et al. reported that rutin inhibited poly(ADP-ribose) polymerase, and induce DNA damage in BRCA mutant cells [50]. Additionally, rutin was reported to induce G0/G1 cell cycle arrest and apoptosis in HPV-C33A cervical cancer cells [51]. In addition, a study showed that treatment of HT-29 human colon cancer cells with hyperoside and rutin of *Nelumbo nucifera* resulted in stimulating mitochondrial apoptosis through a caspase-dependent mechanism [52]. So these mechanisms are expected to play a role in cancer management of bran extract.

In this study, there was a reduction in tumor size when bran extracts were used in vivo for both concentrations (10% bran and 20% bran), with a percentage of change of 50.1% and 82.6%, respectively. These results may be explained by many studies that reported the use of rice bran in vivo. It was reported that rice bran oil enhanced the antioxygenic ability and protected against oxidative stress when used in Albino rats [53]. Additionally, it was noticed that MGN-3/Biobran, which is a rice bran supplement, stimulated apoptosis and immune modulation, resulted in an extremely significant reduction in tumor volume (63.27%) and tumor weight (45.2%) when used in Ehrlich carcinoma-bearing mice [54]. In addition, Choi et al. noticed that when black and brown rice are used to feed mice bearing colon carcinoma cells (CT-26-), tumor growth was inhibited by 35% and 19% in the black and brown bran-fed groups, respectively [55]. Additionally, the treatment lowered VEGF, COX-2, and 5-LOX expression. So these mechanisms may explain the anticancer effect of bran extract when used in vivo.

### 3.2. Immunomodulatory Effect

The immune system is an active system that consists of cells, tissues, proteins, and organs [56]. The immune system can be classified into two subsystems—the innate immune system and the adaptive immune system. The innate system is composed of physical barriers such as skin and immune cells such as macrophages. The adaptive system consists of T lymphocytes (T cells) and B lymphocytes (B cells) [57]. The function of the immune system is to regulate the progression and evolution of the efficient lymphocytes, so they identify and rapidly respond to non-self antigens [58]. Immune system modulation can enhance the effectiveness of anticancer treatment and improve toxicity in normal cells [59]. However, any variations in the immune system are considered immunomodulation, which includes expression, induction, inhibition, or amplification of the immune system components [60]. Lymphocytes are an indispensable constituent of the adaptive immune system and their proliferation is crucial for functional immunity response [61]. In our study, methanol extract showed the highest stimulation index in spleen lymphocytes in the presence of mitogens (Con A or LPS) followed by ethanol extract and n-hexane. In the absence of mitogens, splenic lymphocytes were stimulated equally by ethanol and methanol extract. The previous effect is mainly due to bran’s components as some of them can stimulate lymphocyte proliferation. A study demonstrated that vanillic acid improved the activity of lymphocytes and the release of interferon IFN-γ [62]. Mudgal et al. reported that caffeic acid enhanced immunity in mice by affecting IL-6 and TNF-α cytokines [63]. According to an in vitro study, chlorogenic acid exerted an immunostimulating effect. Chlorogenic acid amplified T-cell activity and decreased tumor development, leading to immunosuppressive factor inhibition [64,65]. Further, ferulic acid exerted an immunostimulating effect [66]. 

To investigate the effect of rice bran on innate immunity, we studied the activity of mouse peritoneal macrophages after treatment with different cytokines. Our findings showed an enhancement in the phagocytic activity of treated murine macrophages. Ethanol extract had the highest phagocytic index followed by n-hexane and methanol extract. Clarification of such results is related to rice bran phytochemicals that exert an immunomodulation activity. For instance, caffeic acid activated murine peritoneal macrophages by improving their lysosomal enzyme activity [67]. Another study revealed that caffeic acid, ferulic acid, and p-coumaric acid augmented the release of ROS, NO, and cytokine (IL-6, IL-1β, and TNF-α) in RAW 264.7 murine macrophage cells, indicating immunostimulatory activity [68]. Furthermore, an aqueous extract of wheatgrass, which is mainly composed of rutin and ferulic acid, had an immunostimulatory effect. This effect is due to the enhancement of nitrite release in macrophage cells and activation of TNF-α, IL-6, and IL-1β cytokines in the monocytic THP1 cell lines [69,70]. Moreover, IFN-γ secretion was significantly increased by rutin, indicating activation of the immune system [37]. Mencherini et al. confirmed that apiin, which is a major component of rice bran, remarkably inhibited the release of NO2 by lipopolysaccharide-activated J774.A1 macrophages [38]. Eventually, to examine the effect of bran extracts on cytokines of murine splenic lymphocytes, we tested IL-2, IL-4, IL-10, and INF-γ cytokines using the TH1/TH2 assay. Needless to say, cytokines are mainly produced by helper T cells, which are subdivided into Th1 and Th2 depending on their function and the type of cytokines they secreted [71]. Th1 cytokines include Interferon gamma and IL-2 and are responsible for protecting the body from intracellular pathogens and cancerous cells. On the other hand, Th2 cytokines are interleukins 4, 5, 10, and 13 and they are crucial in atopy and allergic inflammation [72,73,74,75]. In cancer patients, the Th1/Th2 balance is altered and shifted toward the Th2 response. Many studies revealed that cancer patients have high levels of Th2 cytokines in comparison to Th1 cytokines [76,77]. According to that, we aim to reduce Th2 cytokines levels and increase Th1 cytokines when treating cancer. Interestingly, n-hexane extract decreased IL-4 levels in lymphocytes. On the other hand, methanol extract showed an increase in IL-2 expression followed by ethanol extract. Additionally, ethanol extract increased INF-γ compared to the negative control. For the IL-10 assay, ethanol extract exhibited a minor modulation in its expression. These results might be due to rice bran’s content which modulates cytokines secretion. For example, date seeds which consist mainly of rutin, caffeic acid, vanillic acid, coumaric acid, and gallic acid exhibited an immunostimulating effect. They augmented IFN-γ and IL-2 synthesis while reducing NO [78,79,80]. Additionally, ferulic acid boosts the T helper type 1 immune response in mice by increasing IL-2 levels [81]. Taken together, we can conclude that the various components of rice bran worked together synergistically to stimulate both the innate and the adaptive immune systems and is able to shift the immune response toward Th1 which exert a cytotoxic effect. 

### 3.3. Antibacterial and Antioxidant Activity 

The antibacterial effect of rice bran was evaluated against different bacterial models, Gram-positive, -negative, and -negative resistant *P. aeruginosa*.

Both ethanol and methanol extracts were capable of decreasing the growth of *E. coli*. In a membrane model study, it was shown that flavonoids, especially kaempferol, triggered cell membrane damage in *E. coli*, and this may explain why alcoholic extracts of rice bran had better antimicrobial activity against *E. coli* as they were richer in this flavonoid [82]. In addition, in another study, it was reported that caffeic acid has antibacterial action, as with *E. coli*, it showed a synergistic impact when combined with Imipenem, decreasing the MIC from 2500 to 1574 mg/mL [83].

The ethanol extract reduced the bacterial growth of *B. subtilis.* In a study, different flavonoids were examined for antibacterial activity and rutin was included. The study demonstrated that rutin was effective against Gram-positive bacteria, particularly *B. subtilis* with MIC (16 μg/mL) [84]. Additionally, n-hexane extract exerted antibacterial activity against *B. subtilis.* However, all the extracts exhibited no activity against *P. aeruginosa* at the maximum selected concentration.

According to previous studies, rice bran has shown antioxidant potential [85,86,87,88]. Arab et al. have reported the antioxidant activity of rice bran methanol extract. The results exhibited a 93.9% of DPPH free radical scavenging effect at 50 mg/mL [87]. Further, rice germ of black rice and red rice have shown antioxidant activity at a concentration of approximately 1 mg/mL [88]. 

## 4. Materials and Methods

### 4.1. Rice Bran Supply and Extracts Preparation

Rice bran was provided by local mills in Al-Najaf (south of Iraq). Maceration was used to prepare different extracts using solvents with various polarities. The bran was soaked (100 g/L) in ethanol (70%), methanol (70%), n-hexane, and water for 14 days with continuous stirring. For the water solvent, soaking was carried out for only 24 h to avoid microbial growth. Following that, the supernatant was collected and filtered, then completely dried using a rotary evaporator. The resulting crude extracts were stored in −20 °C until used. All extracts (excluding aqueous extract) were dissolved in DMSO and diluted using a culture medium so that the final concentration of DMSO did not exceed 1%. Tissue culture medium containing 1% DMSO was used as a negative control

### 4.2. Quantitative Analysis of Rice Bran Extracts by Liquid Chromatography–Mass Spectrometry (LC–MS)

Samples have been prepared by dissolving them in 2 mL dimethyl sulfoxide and completed the volume to 50 mL with acetonitrile. Samples were centrifuged at 4000 rpm for 2 min, then 1 mL has been taken to the autosampler. The injection volume was 3 μL. The test was carried out using the Burker Daltonik (Berman, Germany) impact II ESI-Q-TOF system equipped with the Burker Dalotonik Elute UPLC system (Beremen, Germany). The apparatus was activated using the Ion Source Apollo II ion funnel electrospray source (capillary voltage, 2500 v; nebulizer gas, 2 bar; dry gas flow, 8 L/min; dry temperature, 200 °C; mass accuracy, less than 1 ppm; mass resolution, 50,000 FRS; the TOF repetition rate, 20 kHz). Chromatographic separation was performed using Burker solo 2-C-18 UHPLC column (100 mm × 2.1 mm × 2 μm) at a flow rate of 0.51 mL/min and a column temperature of 40 °C. All standards were used for identification of ms/z and the retention time.

### 4.3. Cell Lines and Cell Culturing Condition 

The antiproliferative activity of bran extracts was tested on four cell lines—three of them are human cancer cells and one is the normal cell line. MCF-7 and T47D are human epithelial breast cancer cell lines, while MDA-MB-231 is human hormone-independent breast cancer cells. EMT6/P and fibroblast are the mouse epithelial breast cancer cell line and human skin fibroblast cells, respectively. To achieve a successful cell culture, many factors were followed such as using completed medium and incubating cells in 5% CO_2_ at 37 °C. Based on the type of the cells, the type of culturing medium varied. For MCF-7 and T47D, complete RPMI 1640 medium (PAN-biotech, Aidenbach, Germany) was used, while complete MEM medium (PAN-biotech, Aidenbach, Germany) was used for culturing EMT6/P. High-glucose DMEM medium (PAN-biotech, Aidenbach, Germany) was utilized for MDA-MB-231 and fibroblast cell lines. In this context, a complete culture medium was prepared through adding the following supplements with the required percentage of each type of tissue culture medium. The supplements are: 1% L-glutamine (Sigma, St. Louis, MO, USA), 10% fetal bovine serum (Gibco, Brough, UK), 1% penicillin-streptomycin (Sigma, St. Louis, MO, USA), 0.1% non-essential amino acids (Sigma, St. Louis, MO, USA) and 0.1% gentamycin solution (Sigma, St. Louis, MO, USA).

### 4.4. Experimental Animals

BALB/C mice (age 4–6 weeks) with a body weight of 23–25 g were used. The mice were kept in an animal room, provided with continuous air ventilation and the humidity between 50 and 60% at 25 °C. All the steps of the experiment were carried out based on the standard ethical guidelines and on obtaining the approval of the Research and Ethical Committee at the faculty of Pharmacy Applied Science Private University (Approval Number:2015-PHA-05). The total number of mice used in this study was 35. 

### 4.5. MTT Cell Viability Assay

The different cell lines were prepared and cell viability was checked using trypan blue stain. Then, the cells were seeded in 96-well plate at a density of 15,000 cells per well in complete medium. Following 24 h of incubation, the medium in each well was discarded and the attached cells were treated in triplicate with rice bran extracts. At the beginning, the extracts were dissolved in 0.1% DMSO (AZ Chemo, Ontario, Canada), then a serial dilution method was used to achieve graduate concentrations from 5 to 0.03 mg/mL. After 48 h incubation, the old medium was removed and replaced with 100 µL fresh medium in each well. Then, 20 µL of MTT [3-(4,5-dimethylthiazol-2-yl)-2,5-diphenyltetrazolium bromide] solution (Sigma, St. Louis, MO, USA) was added, and incubated for 3 h at 37 °C and 5% CO_2_. The developed formazan particles were dissolved using DMSO (100 µL/well), and incubated for 1 h. The absorbance was measured using a microplate reader at 550 nm. The percentage of survival was also estimated and compared to the control results (untreated cells). Further, the IC_50_ value was determined using non-linear regression in SPSS.
Percentage of Cell Viability (%) = (OD of treated cell/OD of control cell) ∗ 100

### 4.6. Apoptosis Assay

For this assay, a T47D cell line was chosen depending on the results of the MTT assay and the IC_50_ values. The cells were cultured in 25 cm² tissue culture flasks at a density of 15,000 cells/mL of complete RPMI medium. Following 24 h of incubation, the old medium was removed and the attached cells were treated with rice bran extracts with the following concentrations: ethanol extract (0.4 mg/mL), n-hexane extract (0.3 mg/mL), methanol extract (1.2 mg/mL). The treated cells were incubated for 48 h at 37 °C. After incubation, the medium was discarded and the attached cells were collected and centrifuged. The formed cell pellet was lysed using lysis buffer followed with different steps according to the instructions of the caspase-3 assay kit (Catalogue no. SEA626Hu) (CLOUD-CLONE CORP, Katy, TX, USA). The absorbance was measured at 405 nm using a microplate reader. Fold increase in caspase-3 concentration was calculated compared to the negative control results.

### 4.7. VEGF Assay

T47D cells were seeded at a concentration of 15,000 cells/mL in 25 cm² tissue culture flasks. After 24 h of incubation, the culture medium was discarded and replaced with fresh medium, obtaining rice bran extracts with the following concentration: ethanol extract (0.4 mg/mL), n-hexane extract (0.3 mg/mL), and methanol extract (1.2 mg/mL). Following 48 h of incubation, the supernatant was transferred into sterile tubes and VEGF level was measured using a VEGFA ELISA kit (Catalogue no. SEA143Hu) (CLOUD-CLONE CORP, Katy, TX, USA). The standard curve was estimated using a human VEGFA standard at different concentrations. The result was expressed as the equivalent of VEGFA (pg/mL) for each milliliter of each extract. 

### 4.8. Preparation of the Experimental Diet for In Vivo Assay

The experimental diet was prepared by applying two bran percentages—the first group was 900 g of mouse fodder mixed with 100 g of rice bran, while the second group was 800 g of mouse fodder mixed with 200 g of rice bran. Both components, the bran and the mouse fodder, were separately weighed and grinded. They were mixed based on the mentioned proportions and the dough texture was made by adding water gradually. The dough was clipped and formed into small discs similar to the mouse fodder, and then sun-dried and kept in −20 °C freezer until use (Figure 10).

### 4.9. Tumor Prophylaxis Treatment

Thirty female Balb/C mice were divided into three groups:

Group 1 (n = 10): 10% group (mouse fodder mixed with 10% rice bran),

Group 2 (n = 10): 20% group (mouse fodder mixed with 20% rice bran), and

Group 3 (n = 10): control group (fed normal mouse fodder).

The prophylaxis treatment started two weeks before tumor inoculation. On the third week of the experiment, EMT6/P cells (1 × 10⁶ cells/mL) were prepared and injected subcutaneously in the abdominal area of each mouse. Following 14 days after inoculation, the dimensions of the grown tumors were measured, and then tumor volume was estimated based on the following formula: (A × B² × 0.5), where, A = the length of the longest aspect of the tumor, and B = the length of the perpendicular to A.

In general, the in vivo model was divided into two stages: prophylactic (2 weeks) and treatment (2 weeks).

### 4.10. Preparation of Murine Splenocytes

The spleen was extracted from a Balb/C mouse and ground under sterilized conditions. A cell suspension of spleen cells was prepared using RPMI-1640 medium. After centrifugation, the cell pellets were re-suspended in 5 mL of red blood cells lysis buffer (1 mol/L NH4Cl), with gentle pipetting many times. The suspension was centrifuged for 8 min at 2000× *g* rpm and 4 °C. RPMI-1640 medium was used to re-suspend the splenocytes cells. The splenocytes were ready to be counted and seeded for the study assays. Three Balb/C mice were used in this preparation.

### 4.11. The Lymphocyte Proliferation Assay

The spleen was extracted from a Balb/C mouse, and grinded into small pieces suspended in RPMI-1640 medium. Following centrifugation at 2000 rpm and 4 °C for 8 min, the supernatant was discarded and 5 mL of RBC cell lysis was used to eliminate the red blood cells. This step involved gentle pipetting for several times, and then centrifuge for 8 min. The pellet was re-suspended with a tissue culture medium and was ready to be counted and seeded for the immune assay.

### 4.12. The Lymphocyte Proliferation Assay

In the presence of Con A or LPS

The procedure of this assay based on using MTT [3-(4,5-dimethylthiazol-2-yl)-2,5-diphenyltetrazolium bromide] solution (5 mg/mL). In particular, the prepared splenocyte suspension was seeded onto 96-well plates (in the presence of 5 µg/mL of Con A (Santa Cruz Biotechnology, Santa Cruz, CA, USA) or 4 µg/mL of LPS (Sigma) after determining the density of the cells (2 × 10⁶ cells/mL). Afterward, the cells were treated with bran extracts (5-0.625 mg/mL) and incubated for 48 h at 37 °C and 5% CO_2_. After the incubation period, MTT solution was added (10 µL/well) and incubated for 3 h. The reaction outcome was developing formazan crystals, which were dissolved by adding 100 µL of DMSO solvent. An ELISA microplate reader was used to measure the absorbance at 550 nm. The results were described as the stimulation index compared with the negative control (cells with no treatment).

In the absence of Con A or LPS

The previous steps were followed excluding Con A or LPS addition.

### 4.13. Isolation of Murine Peritoneal Macrophage

Three days before peritoneal macrophage (PEM) harvesting, Balb/C mice were IP injected with 5 mL of 3% (*w*/*v*) brewer thioglycollate medium to stimulate macrophage production. Ice-cold sterile phosphate-buffered saline (PBS) (pH 7.4) was utilized to draw out peritoneal macrophages by injecting the PBS into the mouse cavity and recollecting the fluid containing the cells. After centrifugation, the formed pellet was re-suspended in a complete RPMI-1640 medium.

### 4.14. In Vitro Phagocytic Assay (Nitro Blue Tetrazolium (NBT) Reduction Test)

The nitro blue tetrazolium (NBT) reduction assay was carried out based on Rainard research [89]. Briefly, peritoneal macrophages (5 × 10^6^ cells/well) were seeded onto a 96-well plate along with different concentrations of rice bran extracts (5-0.625 mg/mL). The plate was incubated for 48 h at 37 °C. Then, each well was treated with 20 μL yeast suspension (Astrico, Amman, Jordan) (5 × 10⁶ cells/well in PBS) and 20 μL nitro blue tetrazolium (NBT) (ASTATECH, Bristol, PA, USA) (1.5 mg/mL in PBS), except the negative control wells, which received 20 µL PBS and 20 µL DMSO. After incubation for 1 h at 37 °C, the supernatant was discarded, and the attached macrophages were washed with RPMI 1640. The cells were air-dried before 120 μL 2M KOH and 140 μL DMSO were added to each well. The absorbance was measured at 570 nm using an ELISA microplate reader. The percentage of NBT reduction (phagocytic activity) was calculated according to the following equation [90].
Phagocytic index = (OD sample − OD control)/OD control ∗ 100

### 4.15. TH1/TH2 Assay

In this assay, cytokines (IFN-γ, IL-2, IL-4, and IL-10) were measured for murine splenocytes cultured for 48 h with various extracts of rice bran using mouse TH1/TH2 ELISA kit. In particular, the cell suspension (splenocytes in RPMI-1640 medium) was seeded onto a 96-well plate at a concentration of 2 × 10⁶ cells/mL, along with 100 µL/well of different rice bran extracts (5 mg/mL). Following 48 h of incubation, culture supernatants were isolated to measure the level of cytokines according to the technical procedure mentioned in TH1/TH2 ELISA kit (Catalogue no. 88-7711-44) (Thermo Fisher Scientific, Oxford, UK). To estimate the concentration of each cytokine, a standard curve was prepared for each one. The measured absorbance values were converted into concentration (pg/mL) using standard curve.

### 4.16. Microbial Strains

The microbial strains used in this study were *Escherichia coli* (ATCC 10801) (university Blvd. Manassas), *Pseudomonas aeruginosa* (ATCC 27853), and *Bacillus spizizenii* (ATCC 27853) (KwikSTIK, Grenoble, France). Bacterial strains were transferred onto a nutrient agar slant at 4 °C.

### 4.17. Antibacterial Assay

The minimum inhibitory concentration (MIC) of the rice bran extracts was determined using the microtiter plate dilution method. Concisely, the different extracts were dissolved in Muller–Hinton broth (MHB) using DMSO solvent (0.1%), and then serial dilutions were applied starting with a concentration of 150 to 1.17 mg/mL. The bacteria were suspended in MBH and adjusted to turbidity equal to 0.5 McFarland standards. The microplates were inoculated with bacterial suspension (100 µL/well) and incubated for 24 h at 37 °C. A microplate reader was used to measure the turbidity at a wavelength of 600 nm [91]. The MIC value was determined as the lowest concentration of the extracts at which bacterial growth is prevented. 

### 4.18. DPPH Radical Scavenging Assay 

The assay was conducted using 2,2-diphenyl-1-picrylhydrazyl (DPPH) radical (ChemCruz, Santa Cruz, CA, USA) to evaluate the radical-scavenging activity of rice bran extracts by following the method of Brand Williams [92]. Briefly, the rice bran extracts and ascorbic acid (the reference standard) were prepared at different concentrations (200-12.5 µg/mL) by dissolving them in methanol and distributing them into test tubes. After preparing 0.004% *w*/*v* DPPH solution using methanol, 3 mL of this solution was mixed with 2 mL of the extracts solution and left in a dark place for 30 min. Then, the absorbance was measured using (UV–vis) spectrophotometry at a wavelength of 517 nm. The percentage of scavenging activity was estimated according to the following equation: Inhibition (%) = [(A0 − A1)/A0] 100, where A0 is the absorbance of the control (DPPH solution without any extract or reference compounds), and A1 is the absorbance of the sample. The IC_50_ value (half maximal inhibitory concentration) was calculated using linear regression analysis.

## 5. Conclusions

Rice bran is produced in high quantities without efficient use in food products, especially in the Middle East. Rice bran has the capacity to fight cancer by multiple mechanisms including apoptosis induction, angiogenesis inhibition, and immune system activation. Phytochemical analysis of rice bran revealed the presence of active phytochemicals that may explain its biological activity. Further studies are needed to evaluate the potential of rice bran as a food supplement, with more insight into the mechanisms of action of rice bran components.

## Figures and Tables

**Figure 1 pharmaceuticals-15-01502-f001:**
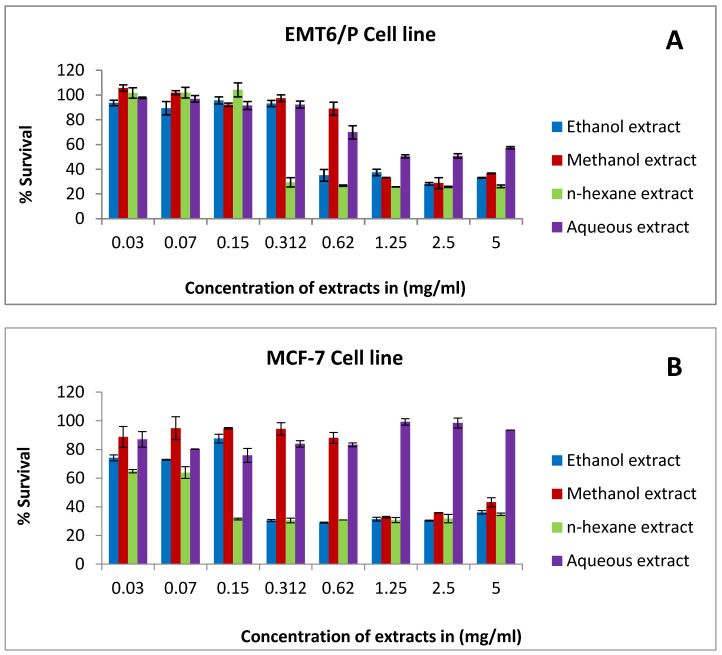
The antiproliferative activity of rice bran extract against: (**A**) the EMTP cell line, (**B**) the MCF-7 cell line, (**C**) the T47D cell line, (**D**) the MDA-MB-231 cell line, and (**E**) the fibroblast cell line using concentrations between 0.03 and 5 mg/mL. Percentage of cell viability (%) was calculated as (OD of treated cells/OD of control cells × 100). Results are expressed as the means of two independent experiments (bars) ± SEM (lines).

**Figure 2 pharmaceuticals-15-01502-f002:**
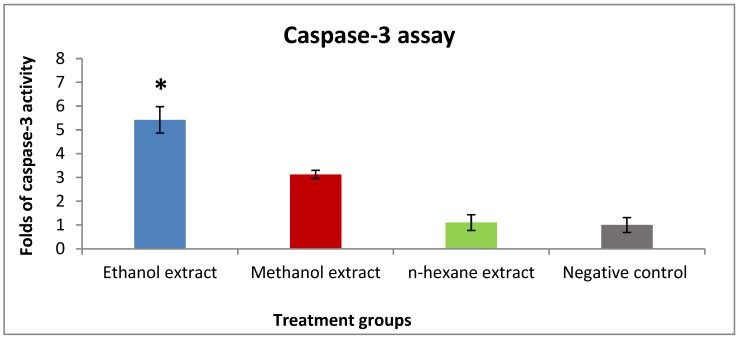
The effect of IC_50_ concentration of rice bran extracts on caspase-3 expression in the T47D cell line. Concentration of the extracts: ethanol (0.4 mg/mL), methanol (1.2 mg/mL), and *n*-hexane (0.3 mg/mL). Results are expressed as the means of three independent experiments (bars) ± SEM (lines) (* *p* value = 0.006).

**Figure 3 pharmaceuticals-15-01502-f003:**
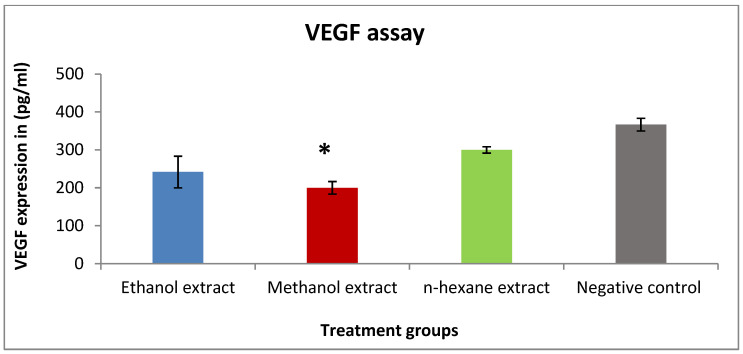
The effect of IC_50_ concentration of rice bran extracts on VEGF expression in the T47D cell line. Concentration of the extracts: ethanol (0.4 mg/mL), methanol (1.2 mg/mL), and *n*-hexane (0.3 mg/mL). Results are expressed as the means of three independent experiments (bars) ± SEM (lines) (* *p* value = 0.04).

**Figure 4 pharmaceuticals-15-01502-f004:**
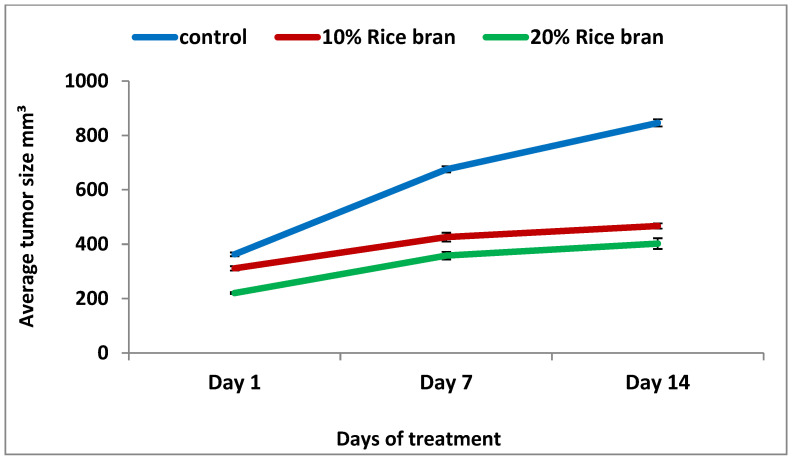
A plot of change in average tumor size (mm³) versus time in days of measurements in tumor-bearing mice.

**Figure 5 pharmaceuticals-15-01502-f005:**
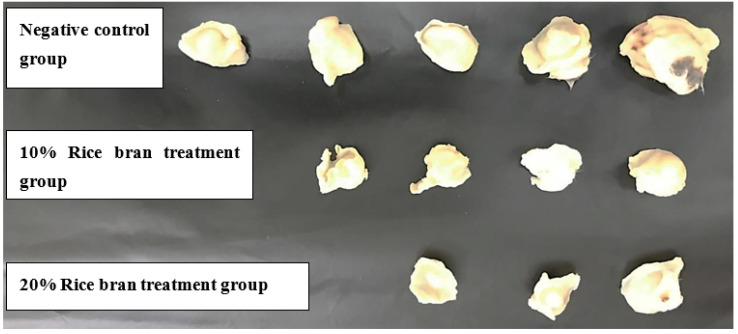
Effect of rice bran consumption on tumor size and cure percentages. The 20% group showed 70% non-detectable tumors compared to the negative control (50%).

**Figure 6 pharmaceuticals-15-01502-f006:**
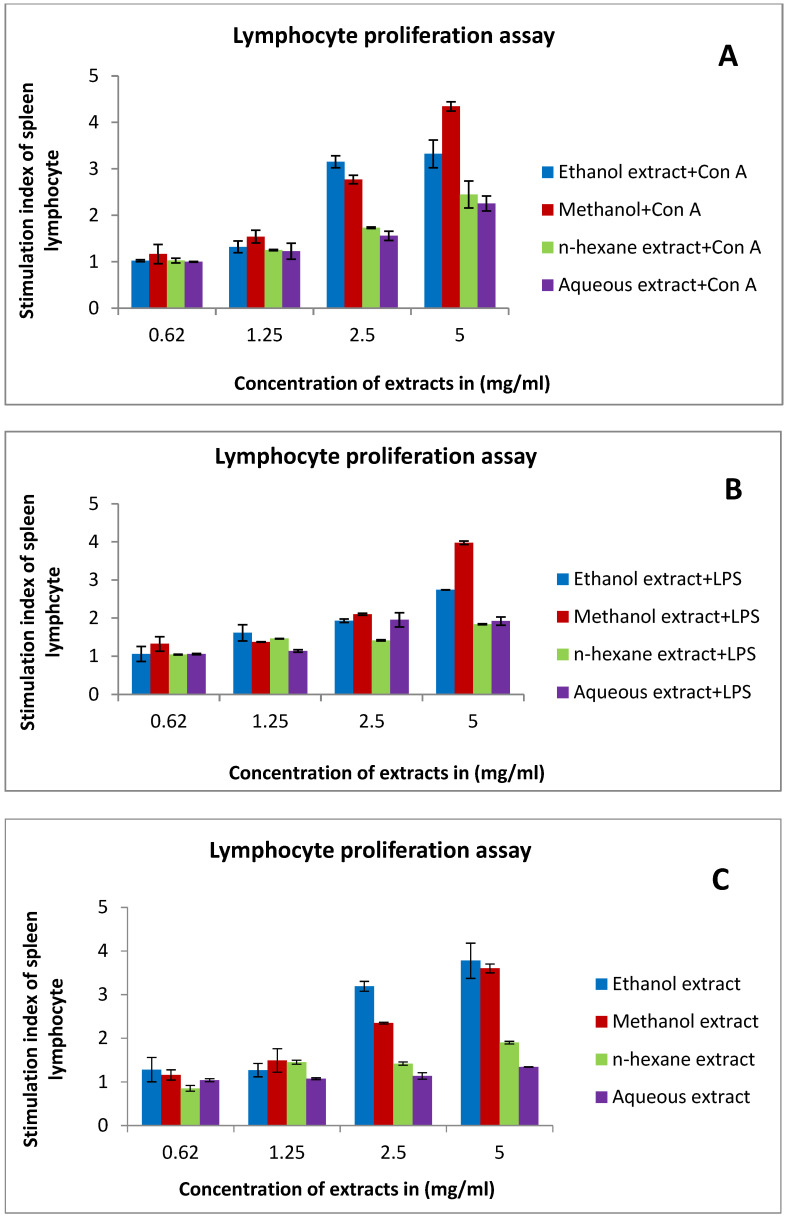
The effect of rice bran extracts on the proliferation of splenic lymphocytes using different concentrations (5–0.62 mg/mL) in the presence and absence of co-mitogens (Con A or LPS): (**A**) in the presence of 5 μg/mL Con A; (**B**) in the presence of 4 μg/mL LPS; (**C**) in the absence of co-mitogens. Results are expressed as the means (bars) ± SEM (lines).

**Figure 7 pharmaceuticals-15-01502-f007:**
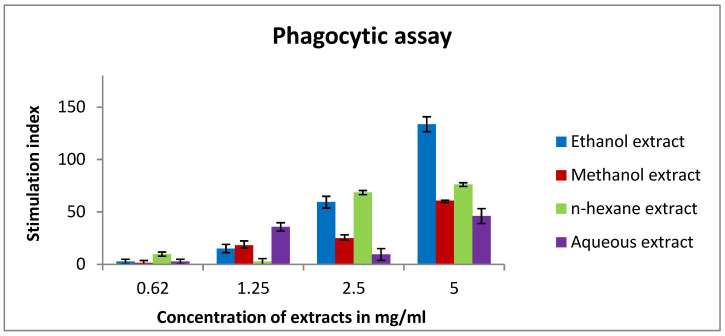
In vitro phagocytic assay using the nitro blue tetrazolium (NBT) reduction test of peritoneal macrophages treated with various concentrations (0.62–5 mg/mL) of rice bran extracts. Ethanol extract had the highest phagocytic index (133). Results are expressed as the means (bars) ± SEM (lines).

**Figure 8 pharmaceuticals-15-01502-f008:**
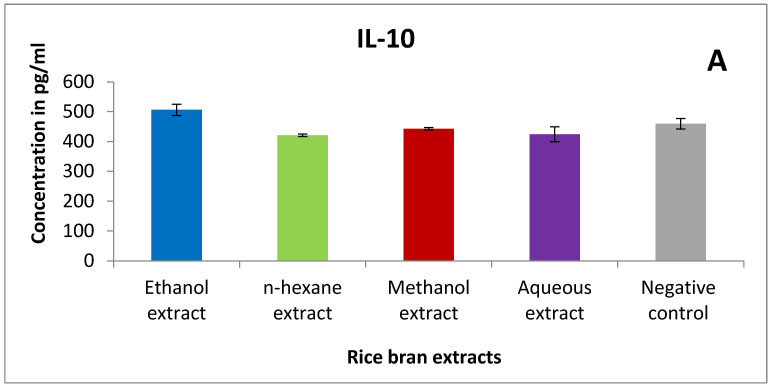
The effect of rice bran extracts (at a concentration of 5 mg/mL) on the expression level of different cytokines (pg/mL). (**A**) IL-10 (**B**) IL-4 (**C**) IL-2 (**D**) IFN-γ. Results are expressed as the means of three independent experiments (bars) ± SEM (lines).

**Figure 9 pharmaceuticals-15-01502-f009:**
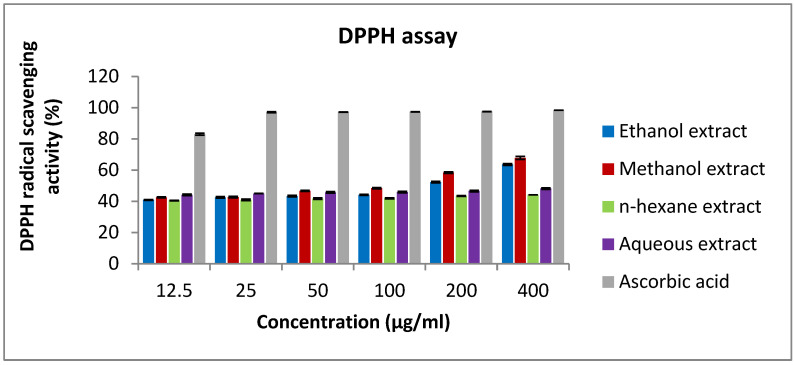
DPPH radical scavenging capacity of rice bran extracts. Methanol extract has the highest percentage of DPPH reduction compared to the other extracts. Values are the means of duplicate analysis (bars) ± SEM (lines).

**Figure 10 pharmaceuticals-15-01502-f010:**
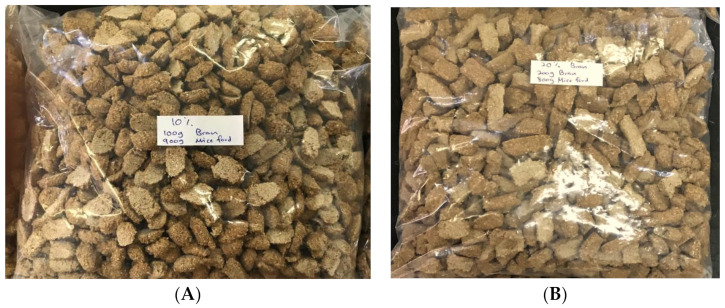
Experimental diet for in vivo study: (**A**) 10% bran (900 g of mouse fodder mixed with 100 g of bran); (**B**) 20% bran (800 g of mouse fodder mixed with 100 g of bran).

**Table 1 pharmaceuticals-15-01502-t001:** LC–MS analysis of rice bran extracts.

NO	Compounds	Formula	RT (Retention Time)	Relative % (Aqueous Extract)	Relative % (Methanol Extract)	Relative % (n-Hexane Extract)	Relative % (Ethanol Extract)
1	Succinic acid	C4H6O4	0.98	70.254	2.805	0.007	0.966
2	Gallic acid	C7H6O5	1.04	0	0.029	0	0.023
3	Protocatechuic aldehyde	C7H6O3	2.09	0.029	0.386	0	0.208
4	Vanillic acid	C8H8O4	2.62	1.318	1.414	0	0.733
5	Chlorogenic acid	C16H18O9	2.88	0	0.008	0	0
6	Vanillic acid	C8H8O4	3.2	0.163	0.784	0	0.367
7	Caffeic acid	C9H8O4	3.27	0.150	2.873	0	1.533
8	Anthranilic acid	C7H7NO2	3.88	0	0	0	0
9	Anthranilic acid	C7H7NO2	4.07	0	1.120	0	3.155
10	p-Coumaric acid	C9H8O3	4.44	0	5.307	0.001	1.841
11	Ferulic acid (trans)	C10H10O4	4.55	0.183	0.266	0	0.075
12	Anthranilic acid	C7H7NO2	4.56	0	3.866	0	0
13	3,5-Dimethoxy-4-hydroxyacetophenone	C10H12O4	4.87	5.934	6.771	0.017	2.524
14	Apiin	C26H28O14	5.11	10.173	31.189	0	15.968
15	Ferulic acid (trans)	C10H10O4	5.13	4.024	13.796	0.022	4.860
16	Salicylic acid	C7H6O3	5.13	0.203	0.918	0.013	0.292
17	2,4-Dihydroxyacetophenone	C8H8O3	5.29	0	0.349	0	0.167
18	Saponarin	C27H30O15	5.48	1.149	4.231	0.001	2.024
19	Rutin	C27H30O16	5.58	0.074	3.852	0	2.193
20	3,5-Dimethoxy-4-hydroxyacetophenone	C10H12O4	5.63	0.066	0.017	0	0.036
21	3-Rha-7-Rha quercetin (NMR)	C27H30O15	5.77	0.016	0.445	0	0.120
22	Salicylic acid	C7H6O3	5.78	1.832	1.762	0.001	0.676
23	Spiraeoside	C21H20O12	5.78	0	1.011	0	0.563
24	3-O-Methyl quercetin	C16H12O7	8.8	0	0.263	0	0.201
25	Apigenin	C15H10O5	9.92	0.462	0.803	0	0.545
26	Kaempferol	C15H10O6	10.13	2.738	0.382	0	0.164
27	5,6,4′-Trihydroxy-7,3′-dimethoxyflavone	C17H14O7	10.34	0.764	8.002	0	5.173
28	ISOHAMNETIN	C16H12O7	10.51	0.339	0.452	0	0.067
29	FAHFA 26:2; FAHFA 18:2/8:0; [M-H]-	C26H46O4	29.53	0.075	2.943	49.380	29.837
30	FAHFA 27:2; FAHFA 18:2/9:0; [M-H]-	C27H48O4	29.99	0.045	3.938	50.555	25.676

**Table 2 pharmaceuticals-15-01502-t002:** IC_50_ values of rice bran extracts against different cell lines.

Cell Line	T47D	MCF-7	MDA-MB-231	EMT6/P	Fibroblast
Rice bran extracts	IC_50_ mg/mL ±SEM	IC_50_ mg/mL ±SEM	IC_50_ mg/mL ±SEM	IC_50_ mg/mL ±SEM	IC_50_ mg/mL ±SEM
Ethanol extract	0.37 ± 0.01	0.36 ± 0.01	0.64 ± 0.09	0.71 ± 0.15	>5 mg
Methanol extract	1.01 ± 0.29	1.29 ± 0.04	2.13 ± 0.14	1.19 ± 0.09	>5 mg
n-hexane extract	0.33 ± 0.01	0.12 ± 0.07	0.51 ± 0.03	0.44 ± 0.01	>5 mg
Aqueous extract	>5	>5	>5	>5	>5 mg

**Table 3 pharmaceuticals-15-01502-t003:** The effect of rice bran on tumor size and weight in mice (*n* = 10) (mm³: cubic millimeter).

Treatment Groups (*n* = 10)	Av. Initial Tumor Size (mm³) ± SEM	Av. Final Tumor Size (mm³) ± SEM	% Change in Tumor Size	% of Mice with No Detectable Tumor	Average Tumor Weight (g)
Control	362.8 ± 1.2	846.2 ± 1.3	133.2	50%	1.31
1:10% rice bran	311.2 ± 2.6	467.2 ± 4.9	50.1	60%	0.60
2:20% rice bran	220.2 ± 0.9	402.2 ± 4.3	82.6	70%	0.64

**Table 4 pharmaceuticals-15-01502-t004:** Minimum inhibitory concentration (MIC) in mg/mL of the rice bran extracts. Microbial species: *Eschrishia coli* (*E. coli); Bacillus subtilis* (*B. subtilis*); *Pseudomonas aeruginosa* (*P. aeruginosa*). Pure gentamicin was included for comparative purposes.

Tested Microorganisms	MIC of the Rice Bran Extracts (mg/mL)				MIC of the Positive Control (mg/mL)
	Ethanol Extract	Methanol Extract	n-Hexane Extract	Aqueous Extract	Gentamycin
*E. coli*	150	150	>150	>150	0.012
*P. auriginosa*	>150	>150	>150	>150	0.012
*B. subtilis*	150	>150	150	>150	0.003

**Table 5 pharmaceuticals-15-01502-t005:** DPPH assay results of rice bran extracts.

Rice Bran Extracts	IC_50_ (µg/mL) ± SEM
Ethanol extract	168.7 ± 9
Methanol extract	114.6 ± 13
n-hexane extract	>400
Aqueous extract	233.1 ± 22
Ascorbic acid	1.74 ± 0.2

## Data Availability

Data is available within the article.

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
