# Peer review of "Immunomodulatory, Anticancer, and Antimicrobial Effects of Rice Bran Grown in Iraq: An In Vitro and In Vivo Study"

_pharmaceuticals, 2022, doi:10.3390/ph15121502_

Round 1
Reviewer 1 Report
The manuscript addresses an interesting aspect of the use of rice bran as a health-promoting agent with anticancer and antimicrobial activity. However, the authors should demonstrate more the novelty of their study. What distinguishes rice bran from iraq - do Iraqi conditions benefit activity? Did the authors determine antioxidant activity of the extracts?
There are also some issues that should be improved:
1. et al. - please check in whole manuscript
2. There are some problem with spaces - lines 74, 76, etc.
3. It is not clear why caspase-3 assay and VEGF test were conducted withoud aqueous extract?
4. Line 331 - "treatment"
5. Materials and methods section - please use spaces between values and units
6. References section - please use italics for latin names
Author Response
Thank you very much for you positive feedback and constructive comments that helped us to improve our manuscript. All you comments were considered in the revised manuscript and a detailed response was also provided.
Attached is our response and changes in the manuscript were highlighted using yellow color. we hope that this revised manuscript will meet you expectations.

Reviewer 2 Report
In summary, in this study the Authors describe the effects of rice bran (growing in Iraq) extracts against a panel of cancer cell lines to prevent cancer and stimulates the immune system. However, the study has several conceptual and methodological weak spots.
First of all, as the authors themselves admit, many studies were conducted on multiple rice cultivars. Therefore the Authors must describe in detail what are the unique characteristics of rice bran grown in Iraq that distinguish it from other types of rice bran with already known effects.
From a methodological point of view, there is a serious error, repeated in every in vitro experiments, which compromises the study. Untreated cells are not the right controls. The right controls are cell treated with the corresponding solvents: ethanol (70%), methanol (70%), n-hexane, and water. Due to the way this study is set up it is impossible to say whether the observed effects are due to the rice bran or to the solvents themselves.
Materials and Methods section is described without scientific precision:
- the manufacturers of the various reagents utilized are not indicated.
- lane 411-416. The Authors must explain what they mean for complete medium. RPMI 1640 and MEM medium are not complete media. FBS, antibiotics, glutamine, non essentials amino acids have been added or not? In what percentage?
Author Response
Thank you very much for you positive feedback and constructive comments that helped us to improve our manuscript. All you comments were considered in the revised manuscript and a detailed response was also provided.
Attached is our response and changes in the manuscript were highlighted using yellow color. We hope that this revised manuscript will meet you expectations.

Round 2
Reviewer 1 Report
The authors have adressed all comments and improved the manuscript.
Reviewer 2 Report
The paper has been improved as requested so It can be published in its current form.